# Ovarian Stem Cells (OSCs) from the Cryopreserved Ovarian Cortex: A Potential for Neo-Oogenesis in Women with Cancer-Treatment Related Infertility: A Case Report and a Review of Literature

Erica Silvestris [1],*[iD], Carla Minoia [2], Attilio Guarini [2], Giuseppina Opinto [2][iD], Antonio Negri [2], Miriam Dellino [3][iD], Raffaele Tinelli [4][iD], Gennaro Cormio [5][iD], Angelo Virgilio Paradiso [6][iD] and Giuseppe De Palma [6][iD]

1 Gynecologic Oncology Unit, IRCCS Istituto Tumori "Giovanni Paolo II", 70124 Bari, Italy
2 Haematology Unit, IRCCS Istituto Tumori "Giovanni Paolo II", 70124 Bari, Italy;
  c.minoia@oncologico.bari.it (C.M.); a.guarini@oncologico.bari.it (A.G.); g.opinto@oncologico.bari.it (G.O.);
  a.negri@oncologico.bari.it (A.N.)
3 Department of Obstetrics and Gynecology, "San Paolo" Hospital, 70123 Bari, Italy; miriamdellino@hotmail.it
4 Department of Obstetrics and Gynecology, "Valle d'Itria" Hospital, 74015 Martina Franca, Italy;
  raffaeletinelli@gmail.com
5 Unit of Obstetrics and Gynecology, Department of Biomedical Sciences and Human Oncology, University of
  Bari "Aldo Moro", 70124 Bari, Italy; gennaro.cormio@uniba.it
6 Institutional BioBank, Experimental Oncology and Biobank Management Unit, IRCCS Istituto Tumori
  "Giovanni Paolo II", 70124 Bari, Italy; a.paradiso@oncologico.bari.it (A.V.P.);
  g.depalma@oncologico.bari.it (G.D.P.)
* Correspondence: ericasilvestris85@gmail.com

**Abstract:** Cancer treatment related infertility (CTRI) affects more than one third of young women undergoing anti-cancer protocols, inducing a premature exhaustion of the ovarian reserve. In addition to ovarian suppression by GnRHa, oocyte and cortex cryopreservation has gained interest in patients with estrogen-sensitive tumors for whom the hormonal burst to prompt the multiple follicular growth could provide a further pro-life tumor pulsing. On the other hand, cortex reimplantation implies a few drawbacks due to the unknown consistency of the follicles to be reimplanted or the risk of reintroducing malignant cells. The capability of ovarian stem cells (OCSs) from fresh ovarian cortex fragments to differentiate in vitro to mature oocytes provides a tool to overcome these drawbacks. In fact, since ovarian cortex sampling and cryopreservation is practicable before gonadotoxic treatments, the recruitment of OSCs from defrosted fragments could provide a novel opportunity to verify their suitability to be expanded in vitro as oocyte like cells (OLCs). Here, we describe in very preliminary experiments the consistency of an OSC population from a single cryopreserved ovarian cortex after thawing as well as both their viability and their suitability to be further explored in their property to differentiate in OLCs, thus reinforcing interest in stemness studies in the treatment of female CTRI.

**Keywords:** cancer treatment related infertility (CTRI); cryopreserved ovarian cortex; oocyte like cells (OLCs); ovarian stem cells (OSCs); stem cell differentiation

## 1. Introduction

The incidence of cancer during a woman's procreative life is apparently increasing with age, and it rises to 0.003% in the nearly 45-year aged female population [1]. It is estimated that a number as high as up to 680,000 of new women are annually diagnosed with cancer in the United States and that 10% of them are affected during their reproductive age [2], while since 2008, even the Italian Cancer Register (AIRTUM) declared an annual 1.6% increase of cancer incidence in this population [3]. On the other hand, the ongoing COVID-19 pandemic diffusion dramatically influenced in the past two years the management of these patients with several complications such as delay of diagnoses and follow-up care,

interruption of treatment, and an increase in infection rates, which ultimately resulted in a higher cancer mortality [4]. However, despite the daily troubles in managing our patients, the global oncology community is devoting high attention to COVID-19 pandemic effects by improving proper levels of investigation, collaboration, and innovative technologies as well as by adopting dedicated COVID-19 registries that are today helpful in our understanding of the natural history, risk factors, management, and outcomes of cancer patients diagnosed with COVID-19 [4].

It is well known that at least one third of women under 45 years undergoing anti-cancer protocols including gonadotoxic therapies with alkylating agents, small molecules and radiotherapy, or their combination, often develop irreversible side-effect damage to the reproductive system [5–9], with a premature exhaustion of the ovarian reserve (OR). This pathological condition is defined as 'cancer treatment-related infertility' (CTRI), and it primarily affects more than one third of women following their anti-cancer treatments, resulting in persistent discomfort such as a chronic depressive state when the cancer heals. Particularly in young women, the inability to conceive is a negation of their sexual identity.

A primary approach to prevent ovarian desertification in young patients undergoing anti-cancer protocols was based on GnRH agonist (GnRHa) administration during gonadotoxic treatments to protect the OR [10,11] in relation to the hypothesis that the 'hormonally suppressed ovary', namely a dormant ovary, is poorly damageable by drugs and other oocyte injuries [12,13]. However, despite the hormonal suppression of oogenesis, it provides a putative biological protection to oogonial cells by preventing their maturation; data on the post-treatment pregnancy rate are limited and inadequate to support the validity of this option since the suppressed oogonial cells are equally exposed to gonadotoxic agents during anti-cancer treatments [14–16].

Currently, in young women with cancer, a well-established practice for fertility preservation (FP) is provided by oocyte recruitment and cryopreservation for fertilization after the cancer. However, in the presence of hormone-sensitive cancer in prepuberal age, several concerns related to the possibility to reinforce the cancer activity should advise caution in using this method that, on the other hand, is inappropriate for the scheduled time when anti-cancer treatments are urgently needed. Therefore, further strategies concerning the collection of immature oocytes at the germinal vesicle-stage for a subsequent in-vitro maturation (IVM) and fertilization (IVF) [17–19] as well as the cryopreservation of ovarian cortex strips for autologous transplantation after the cancer healing [20,21] have progressively gained interest by fertologists and gynecologists as alternative options in FP practice. Among these options, the cryopreservation of the ovarian cortex requires less cell manipulation, and its finalization is the replacement of the original tissue in exhausted ovaries. However, despite the simplicity of this method and the biologic safety for not using hormonal stimuli and independence from menstrual cycle phase, there are a few underestimated aspects and risks that prevent an extended utilization of such a procedure. These include the unknown or frequently inadequate pool of mature oocytes resident in cryopreserved cortex strips as well as the viability of eggs after thawing and, ultimately, the described risk in hematologic disorders to reimplant malignant cells originally resident in a frozen ovarian cortex, as for leukemia cells [22]. Therefore, several clinical and basic research investigators are constantly focusing alternative strategies to improve the fertility potential in patients with ovarian exhaustion or reduced OR due to anti-cancer gonadotoxic treatments.

Kawamura and co-workers proposed the in vitro stimulation of biologic activity in primordial multiple follicles (PMF) in the context of a poor OR by upregulating the PI3K/Akt pathway through in vitro administration of molecular inducers to neutralize the disruption of Hippo pathway resulting from ovarian cortex fragmentation [23,24]. However, other studies have not supported the suitability of this method [25,26].

Novel information related to potential application in the FP field in 2004 reported the existence of mitotically active germ cells in postnatal mouse ovaries which was also confirmed in other mammalian species, including humans. While this discovery opens new

avenues for implementing neo-oogenesis and folliculogenesis in adult life and counteracting all forms of ovarian failure [27–30], a few skepticisms raised in relation to a foundational dogma in female mammalian reproductive biology, originally proposed in 1951 [31] and subsequently rediscussed [32–40], which described the fixed consistency of the pool of oogonial progenitors at birth that is not thus subjected to renewal during the female postnatal life. In this regard, other investigators provided further evidence of postnatal oogenesis, namely neo-oogenesis, in mammals by isolating even in women the ovarian stem cells (OSCs), as germ cells responsible for the continuing oocyte regeneration [32,41–49]. Furthermore, the viability of OSCs to undergo oocyte maturation was demonstrated by labeling these cells in mice with a fluorescent reporter to investigate the cell fate tracking and once transplanted into ovaries of adult female mice, OSCs differentiated in mature eggs apparently ready to be fertilized [33].

Additional investigation by Guo and co-workers using genetic mouse models provided more knowledge on the OSC fate in vivo, including the postnatal oogenesis in adult females, and they confirmed the occurrence of germ cell meiotic entry into a common neo-oogenesis program during reproductive life [33,50]. These results have been confirmed from other groups [51] and support the functional properties of OSCs in adult mammalian ovaries showing their capability to differentiate into oocyte like cells (OLCs) and ultimately in mature oocytes even from a woman's ovarian cortex thus overturning the long-standing paradigm of the "fixed ovarian pool" [51].

Employment of OSCs to induce neo-oogenesis in FP programs and particularly in CTRI is thus of great interest since the potential utilization of these cells to induce neo-oogenesis avoids additional oncogenic risk related to folliculogenesis induction. In this context, several studies are in progress to optimize their suitability to undergo mature oocyte differentiation in vitro and subsequent cryopreservation. On the other hand, the possibility to derive OSCs from ovarian cortex strips already cryostored from cancer women waiting for their disease resolution before undergoing FP approaches needs to be investigated to offer alternative options.

Here, we review several aspects of the biologic properties of OSCs along with preliminary data on the viability assessment and the functional patterns of the same cells derived from a cryostored cortex. Though initial, our results suggest that even from cryopreserved fragments of ovarian cortex, it is possible to isolate aliquots of OSCs that could apparently be suitable to differentiate to mature oocytes.

## 2. Ovarian Stem Cells to Regenerate Folliculogenesis

Based on the demonstration of the existence of OSCs in mammalian ovaries, several studies focused their phenotype and their molecular characterization. Original observation in mice described on these cells the expression of Ddx4 (DEAD box polypeptide 4) [52], a germline oogonial marker exposed as the transmembrane protein of 79.3 kDa, whose extracellular domain was lost with oocyte differentiation, whereas its cytoplasmic tale was preserved during the maturation process along with other stemness markers as OCT-4 and further variably expressed proteins including Fragilis, Stella, and SSEA-4 [53]. Identification of this marker allowed for the isolation of OSCs from murine and human ovaries through immunoadsorption and flowcytometry sorting [32], and their separation allowed their molecular characterization and culture to investigate their functional properties. By using specifically enriched media, cultured OSCs differentiated in large OLC as apparently mature oocytes expressing specific genes of final differentiation as GDF-9 and SYCP3 [32] and capable of completing their meiotic process through haploidy acquirement. This status has been assessed by analysis of their DNA content [54] as well as by FISH of chromosome number showing single signals on chromosome X and 5 [32]. By using prepuberal female mice, Telfer and co-workers demonstrated that after sterilizing these animals by busulfan, the intra-ovarian reimplantation of OSCs resulted in definite neo-oogenesis with oocyte repopulation while the subsequent fertilization of these animals produced favorable outcomes in pregnancy [55].

However, there is an open debate on the realistic existence of OSC in mammalian ovaries. To this regard, several investigators claim the role of the Ddx4 molecule as a marker of oogonial cells [56], as well as its specific expression in both human and mouse ovaries [57], whereas recent observation by Wagner and co-workers described that ovarian perivascular mesenchymal cells are detectable by anti-Ddx4 reagent while OSCs are undetectable in mammalian ovaries [58]. With respect to such a skepticism, the number of studies describing the occurrence of OSCs is perhaps equivalent to the denying reports while their biological properties assessed by independent groups of investigators apparently support that OLCs differentiated by human OSCs are functional cells in vitro though these properties need to be better ascertained for their potential use in vivo.

In this regard, the debate on mammalian neo-oogenesis arose within the scientific community by several studies published by independent groups of investigators, namely Pacchiarrotti J [28], Bukovsky [41], Johnson [42], Virant-Klun [43], Niikura [44], Zou [45], Oktem [46], Parte [47], Zhang [48], and Zhou [49], supporting the existence of SCs in the ovary (germline stem cells (GSCs) [28,41,42,45,48,49], embryonic-like stem cells of the adult [43], premeiotic germ cells [44], and very small embryonic-like putative stem cells (VSELs) [47]), whereas others including Bristol-Gould [34], Liu [35], Byskov [36], Kerr [37], Zhang [38], Lei [39], and Yuan [40], confuted these results on the basis of a technical misunderstanding of data and they required further evidence for the occurrence of OSCs in all mammals, including humans.

In this contest, there was an additional question regarding the use of Ddx4 as a germline marker for sorting OSCs [33], which is considered a functional membrane marker for their isolation, while its location is apparently detected within the cytoplasm. In reality, it has been reported that the Ddx4 transmembrane-spanning domain is expressed by OSCs in their early maturation step before its setting in a definite intracytoplasmic location in mature oocytes [54,59].

On the other hand, Ddx4$^+$OSCs has been clearly demonstrated to express stemness markers such as Oct4 and SSEA4, thus providing evidence in support of active germ cells in an adult mouse ovary [33]. The expression of those stemness markers has been confirmed by several methods including detection by specific reagents, RT-PCR, and BrdU up-take [60]. Similarly, IFITM3 (Fragilis) has also been found to be a highly specific marker that is based on its transmembrane location, and it is functionally utilized for efficient OSC isolation [61]. In fact, Sequeira RC and colleagues in their study proposed a new fruitful isolation method combining anti-Ddx4 and anti-IFITM3 antibodies in MACS sorting that definitively improved the recruitment of OSCs with respect to the method using only santi-Ddx4 antibodies [62].

Nevertheless, several aspects need to be better defined in relation to the obtainment of homogeneously expanded populations of well-differentiated oocytes in vitro. In this regard, an intensive investigation is in progress to identify specific oogenic factors enrolled in the final differentiation stage of mature oocytes. Ongoing research from major investigative groups postulate that undifferentiated granulosa cells, capable of interaction with newly generated oocytes and formation of the primordial follicle, prime the final maturation of OLCs in relation to the expression of stemness markers and genes [1]. However, this aspect is moreover controversial since the undifferentiated granulosa cells probably derive from pluripotent stem cells (PSC), which are resident in ovaries, and under several conditions they can be induced and expanded in vitro [63]. Therefore, in search of a defined model to improve the in vitro maturation process of OSCs to mature oocytes particularly in cancer-associated infertility, several investigators suggest the utilization of patient-derived induced-PSCs to generate autologous granulosa cells capable of inducing aggregation of OSCs and OLCs with the purpose of expanding the folliculogenesis in vitro and selecting the best mature oocytes [64]. A further hypothesis to improve the in vitro differentiative potential of OSCs in mature oocytes concerns the complementary role of cumulus cells as well as the complete assessment of follicular fluid components, which in vivo exert

the most efficient activity to drive the oocyte maturation and several acquisitions are in progress [65].

These results validate the hypothesis that OSCs are capable of regenerating the oogenesis and that they are suitable to be further investigated for a future enrollment in FP programs in women with POF or CTRI. Particularly in young patients with cancer, the utilization of OSCs to promote the neo-oogenesis will be safe and easily affordable since their recruitment and their process to obtain OLC progenies provides the advantage to select the most viable OLCs to be cryopreserved without oncogenic risks due to hormone stimulation with the advantage of selecting the best eggs to be used.

## 3. The Cryostored Ovarian Cortical Strips as a Source of OSCs

Although not highly diffused in the majority of cancer FP centers, the biobank of cryopreserved ovarian tissue provides a considerably source of biological material to investigate the OSC population of women. Here we revisit major aspects useful to obtain OSCs from cryostored strips.

*Cortical ovarian tissue isolation and cryopreservation*—Ovarian tissue cryopreservation is an immediate, alternative FP procedure for female cancer patients urgently requiring oncological treatments. The recovery of cortex strips of approximately 1 cm by 4–5 mm [66] is generally obtainable by laparoscopic biopsy [67] to minimize possible injuries to stromal components and to preserve the strips for the best survival before their freezing and before its cryostorage; the main fragment is commonly divided into small slices of approximatively 0.5–2 mm wide [68]. Suggestion form literature describes two main reasons to process the recovered ovarian tissue fragments and to prepare thin slices, which include the necessity to warranty a better neo-vascularization after reimplantation and to reduce the risk of ischemia when using thicker slices [69], as well as the evidence that the mentioned size of slices appears suitable to a valid permeation of cryoprotective agents in ovarian tissue to prevent dehydration and injuries derived from extreme changes of temperature due to freezing and thawing. However, other studies reported that larger strips of approximately 8–10 mm by 5 mm can also be safely cryopreserved and that once reimplanted they maintain their endocrine function better than smaller pieces, thus suggesting that even in larger pieces the neovascularization is functionally recovered after freezing and thawing [66].

The cryopreservation of ovarian tissue is usually performed by two techniques: the slow freezing and the vitrification, as reported in Figure 1 [70]. Slow freezing is a process to avoid tissue injury for quick temperature variation and uses lower concentrations of cryoprotective agents in a freezing equipment capable to perform programmable sequences of lowering temperature. The tissue is cooled to below the freezing point with or without seeding and then, at a controlled cooling rate, slowly frozen between −35 °C and −130 °C before immersion in liquid nitrogen. Several slow freezing protocols have resulted after auto-reimplantation of thawed ovarian tissue in live birth rates that substantially differ in relation to the cryoprotective agents including DMSO, serum, albumin, propanediol, sucrose, ethylene, or glycol as well as their concentration. However, the best live birth rate, which is estimated as equal to approximately 30% after thawing, was obtained when using DMSO [20,71,72].

The vitrification of the ovarian tissue is considered the most suitable option for the FP in prepubertal girls undergoing gonadotoxic treatments. The procedure utilizes solutions at high viscosity along with high concentrations of cryoprotective agents [73]. Ethylene glycol has been proposed in substitution or in combination with DMSO to reduce the cell toxicity of the cryoprotective agents, and relative results suggest a modest benefit in terms of follicular recovery [74]. Over time, the vitrification has been performed using different protocols and variations of exposure intervals to cryoprotectants as well as their concentrations and permeability potential [75]. To avoid cellular toxicity, the exposure time to these reagents is a parameter to be constantly considered during ovarian tissue vitrification [76], for example a protracted time for immersion of the strips can result in high cryoprotectants' absorption, particularly in vitrification of thin tissue slices [77].

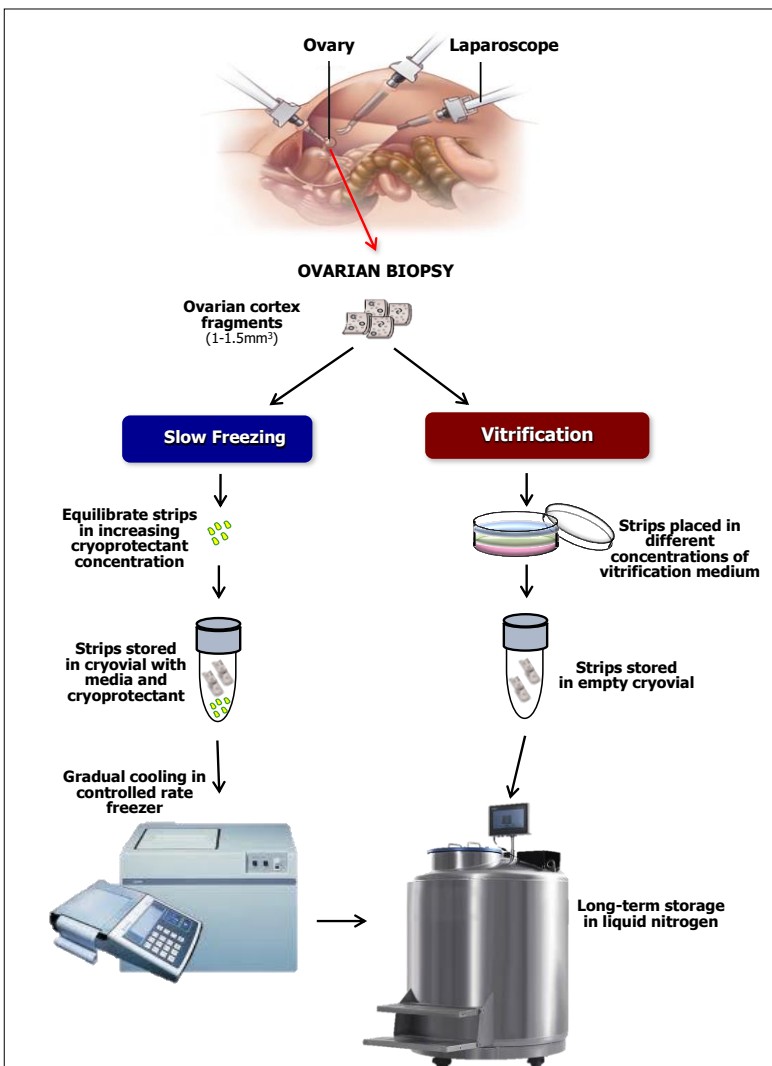

**Figure 1.** Schematic representation of two procedures of ovarian cortex cryopreservation by laparoscopic retrieval.

*Ovarian cortex cryopreservation in cancer patients*—Cryopreservation is indicated for adolescents and women whose cancer treatment is urgently required and cannot be delayed. The first birth obtained after ovarian tissue cryopreservation and auto-reimplantation to a prepubertal patient was reported in 2015 [78]. In a report series from five leading centers in Europe, adopting this procedure in almost 90% of patients with malignant diseases—including hematologic and solid cancers as breast, cervix, borderline ovarian tumors, Ewing sarcoma, and melanoma—the live birth rates were 30% and 21%, respectively, when considering both conceptions after reimplantation and those using IVF of eggs after ovarian tissue thawing [79].

*Drawbacks in using the cortical graft*—Several surgical techniques are currently being used for frozen-thawed ovarian tissue auto-reimplantation. However, to be successful at least two points need to be respected, namely a well vascularized orthotopic transplantation site and joining ovarian tissue by stitches whose edges are fixed with fibrin glue. Suitability of orthotopic sites for auto-transplantation is pivotal for ovarian activity renewal with described efficacy for restoring fertility in over 40% of patients and exponential success rates in terms of pregnancies [80].

Safety issues surrounding ovarian tissue reimplantation from cancer patients have represented a few concerns. Several studies investigated the risks of reintroducing malignant cells together with frozen-thawed ovarian tissue, which could induce a recurrence of

the primary tumor [81]. In the series of patients undergoing ovarian tissue reimplantation described by Dolmans, a percent as high as up to 4.2% of the population showed relapse of the original malignancy [81], thus proving that although preliminary studies describe as suitable this approach in fertility restoration programs, it still appears inadequate in cancer patients for the putative risk of the cancer relapse. In particular, the reintroduction of cancer cells with ovarian tissue transplantation appears as drawback in hematologic malignancies as acute leukemias in which circulating leukemic cells are present in cryopreserved ovarian biopsies that can regenerate the disease after the procedure [82].

Based on these considerations, the cryostorage and reimplantation of ovarian tissue for the FP includes both advantages and drawbacks, particularly in oncology. Whilst it has been improved in technical methodology for acquiring viable follicles as well for improving the grafting of implanted ovarian biopsies, on the other hand the selection of follicles is based only on their ultrasound detection and the quality of oocytes to be reimplanted is usually not investigated. Although the risk to regenerate certain tumors in relation to the presence of cancer cells in the cryopreserved ovarian cortex can be limited by selecting the best time for the biopsy, the procedure cannot be considered completely safe at least in patients with hematologic and a few solid malignancies.

### 4. OSC from Cryopreserved Ovarian Cortical Fragments: A Preliminary Observation

Despite encouraging results on the potential of OSCs from freshly derived strips of ovarian tissue to generate OLCs and their subsequent differentiation to mature eggs, at present very few information is available on both viability and maintenance of biological properties of the same cells after freezing for cryopreservation and thawing as reported for the cortex slices reimplanted in ovaries. This information is particularly needed due to the large availability of cryopreserved cortex specimens in biobanks of institutional centers deputed to FP programs, which could indeed take considerable advantage of deriving mature oocytes from cryostored ovarian samples.

Therefore, here we describe our preliminary experience in verifying both occurrence and viability of OSCs from ovarian cortex fragments after thawing.

During the past two years, we collected ovarian cortex biopsies from 11 non-menopausal women (age 20–43 years) with the purpose of expanding our previous studies on OSCs by verifying their suitability after cryopreservation. The patients recruited were affected by urogynecology tumors with no ovarian involvement and before abdominal surgery, chemo-radiotherapy, with serum anti-Müllerian hormone (AMH) levels higher than 1.4 ng/mL and antral follicle count (AFC) greater than four by ultrasound evaluation. The cortical biopsies were performed mostly by laparoscopy during the abdominal surgery or by specifically needed laparotomy in single instances and all patients agreed to the cortex sampling by dedicated informed consent to donate their biological material for research purposes. Once obtained, the cortex pieces were quickly prepared for the cryopreservation within one hour and their freezing in liquid nitrogen was performed according to the 'slow freezing' protocol [20]. However, due to COVID-19 restrictions in our institution, we were able to process a limited number of samples and here describe the results obtained by a representative case.

The cortex piece was derived from a 28-year-old woman, with AMH 3.56 ng/mL and AFC 28 follicles values, submitted to left annexiectomy for a 7 cm benign ovarian cyst. To perform our test, the ovarian sample was originally maintained in Leibovitz L-15 medium in ice and the medulla was gently scraped by a scalpel to isolate the cortical tissue. Thus, the cortical parts were subsequently fragmented in strips of 5 by 5 by 1–2 mm, then maintained in the same medium added with 4 mg/mL human serum albumin and 1.5 mmol/L DMSO, and ultimately frozen by a controlled slow-freezing procedure and then stored in liquid nitrogen. Thus, after the cryopreservation period, the vials containing the cortical strips were thawed in a warm water bath at 37 °C for 2 min and the strips were then carefully washed three times with fresh medium to remove cryoprotectant.

To better protect cell viability, isolation of OSCs from the strips was performed by an enzyme-free method since energic enzymes as collagenase may damage the cell membrane proteins. Briefly, the strips were further fragmented and then placed in 35μm of disposable disaggregating medium (Medicon, Becton Dickinson, Singapore) containing a phosphate buffer was added by the Medimachine® tissue homogenizer. The recruited cell suspension was then stratified on a Ficoll gradient and centrifuged; then, it was recovered and incubated at 4 °C with rabbit polyclonal anti-Ddx4 (Abcam, ab13840, Dublin, Ireland) followed by further incubation with anti-rabbit IgG-FITC antibody (Sigma-Aldrich, St. Louis, MO, USA, F0382). The OSC suspension was then incubated with 7-amino-actinomycin D labelled with phycoerythrin-5 (7-AAD-PC5 Viability Dye, Beckman Coulter, CA, USA) as viability dye, and it was ultimately evaluated by flow cytometric analyses (NAVIOS® flow cytometer, Beckman Coulter). The analysis was performed in technical triplicates to obtain standardized values.

As shown in Figure 2, by adapting FSC vs. SSC to the OSC parameters, a large population of these cells was detected (Figure 2A). Moreover, by using the 7-AAD vital staining, we found that almost all cells included in this subset, namely 95.7% ± 2.5 of OSCs, were living cells as compared to the subset of those that were dead equal to 4.3% ± 0.8 (Figure 2B). The incorporation of the vital dye was clearly different between the two subsets to support the viability of these cells after their thawing (Figure 2C).

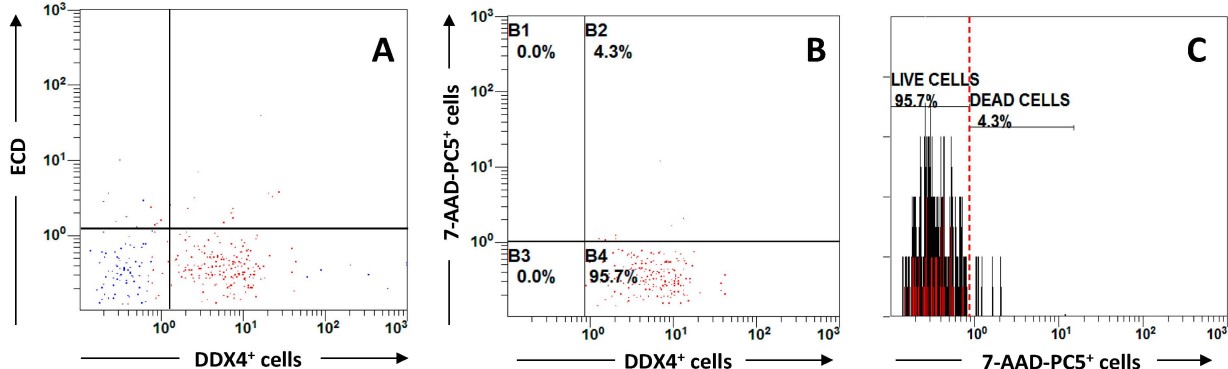

**Figure 2. Flowcytometry analysis of ovarian stem cells (OSCs) isolated from a thawed ovarian cortex sample.** After isolating the cell suspension by Ficoll density gradient, the cell population incubated with FITC-conjugated anti-Ddx4 oogonial marker included a large consistency of positive cells (**A**), which were then analyzed in their vitality using the PC5-conjugated 7-AAD viability marker. As shown, almost the full population, namely 95.7% of Ddx4-positiive cells were found viable in contrast with a minority equal to 4.3% of dead cells (**B,C**). This result suggested that the cryopreservation of ovarian cortex fragments in liquid nitrogen is almost indolent on the OSC viability and that probably other functions including their capability to differentiate in oocytes in vitro is also restored after their thawing. (Values are percentage of positive cells measured on 10,000 events).

Although preliminary and completed on a single sample, our results are in accordance with the viability data of a previous study completed by Wagner [60] who described a similar viability extent of the Ddx4-positive cell population (89%) obtained from a vitrified cortical specimen and thawed before the analysis. To confirm this observation, we plan to increase the number of samples of cryopreserved ovarian cortex to be investigated in the OSC properties to maintain viability and functional attitude to differentiate in oocytes after thawing.

Thus, our results on Ddx4-positive cell viability from thawed strips along with data from Wagner [58], although including preliminary information, at least serve as a quality control of the ovarian cortex cryopreservation process, and they suggest the suitability of both slow-freezing and vitrification procedures in warranting viability in OSCs after their separation from cryopreserved specimens.

## 5. Conclusions

Recent advances in the field of regenerative medicine have focused an increasing interest concerning the employ of stemness technology as a possible novel application in the treatment of cancer-related-infertility for young women at risk for CTRI. Since a number of studies proved the ability of cultured OSCs to differentiate into OLCs, their future utilization is to be necessarily considered a promising practice not only to treat infertility in women affected by POF or to restore the hormonal balance in post-menopausal age, but a safety alternative to oocyte and ovarian cortex cryopreservation in patient candidates for FP programs and particularly in young women at risk for CTRI.

Their major suitability in this field concerns the possibility to design an FP program based on OSC recruitment from fresh ovarian cortex fragments before gonadotoxic treatments, differentiation in mature oocytes, selection of populations of most viable eggs, and cryopreservation. Thus, the main advantage of this approach includes the availability of a large population of oocytes from each patient, the full absence of risks for the estrogenic stimulation or of reintroducing malignant cells, and immediate applicability to pre- and post-pubertal patients.

The possibility to isolate OSCs viable up to 95% from cryostored ovarian cortex, a percentage similar to fresh samples, also suggest that this cellular population is apparen-tly resistant to the temperature stress for freezing and thawing thus maintaining excellent viability and perhaps the capability to generate mature and fertilizable oocytes in vitro. This is exactly what we will explore in a next study by verifying the expression of oocyte molecular markers in cultured OSCs from cryopreserved samples.

**Author Contributions:** E.S. conceptualized the study, wrote the manuscript, and critically reviewed the manuscript; C.M. and A.G. contributed to writing; G.O. and A.N. contributed to experimental work; M.D. and R.T. contributed to writing; G.C. critically reviewed the manuscript, G.D.P. and A.V.P. contributed to experimental work and wrote the manuscript. All authors have read and agreed to the published version of the manuscript.

**Funding:** This work has been supported by a grant (5x1000) from the IRCCS Istituto Tumori "Giovanni Paolo II" Bari, supporting the program exploring ovarian stem cell development to prevent infertility in oncological patients.

**Institutional Review Board Statement:** The study was conducted in accordance with the Declaration of Helsinki, and approved by Ethics committee of IRCCS Istituto Tumori "Giovanni Paolo II" (Del. 332/2019).

**Informed Consent Statement:** Informed consent was obtained from subject involved in the study.

**Data Availability Statement:** Not applicable.

**Conflicts of Interest:** The authors declare no conflict of interest.

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
