# Peer review of "Ovarian Stem Cells (OSCs) from the Cryopreserved Ovarian Cortex: A Potential for Neo-Oogenesis in Women with Cancer-Treatment Related Infertility: A Case Report and a Review of Literature"

_cimb, doi:10.3390/cimb44050157_

Round 1

Reviewer 1 Report

I had some small remarks to the previous version of the manuscript whereas now I'm satisied. All my concerns were clarified. As I already stated in my opinion it is nice paper with very important, underestimated topic. The authors did a good job and all introduced amendments are enough. I recommend publication in its present form.

Author Response

We are grateful for the favorable appreciation of our work.

Reviewer 2 Report

In my first comments, I asked that the authors cite and discuss the very many papers stating that they have never found ovogonial stem cells in the ovaries of their mammalian model. But in the second version of this “review”, the authors contented themselves with citing the articles contradicting the presence of these cells in the ovaries, without discussing them. They just added the following sentences:

However, there is an open debate on the realistic existence of OSC in mammalian ovaries. To this regard, several investigators claim the role of Ddx4 molecule as marker of oogonial cells [56], as well as its specific expression in both human and mouse ovaries [57], whereas recent observation by Wagner and co-workers described that ovarian perivascular mesenchymal cells are detectable by anti-Ddx4 reagent while OSCs are undetectable in mammalian ovaries [58]. With respect to such a skepticism, the number of studies describing the occurrence of OSCs is definitely higher than the confuting observations and evidence from their discovery and assessment of biological properties from independent groups of investigators, is enough to support that oocytes differentiated by human OSCs are functional cells in vitro and that these properties need to be better ascertained for their potential use in vivo.”

These are the only sentences added!

First, I would be very interested to discuss with the authors about a quantification of the number of papers that state that OSC exist in comparison with papers that state the contrary (“…the number of studies describing the occurrence of OSCs is definitely higher than the confuting observations and evidence from their discovery and assessment of biological properties from independent groups of investigators, is enough to….”)! And I would like to know the arguments of the authors when they state that the other observations are “confuting”. For me they are not: all say that OSCs do not exist in mammalian ovaries.

The authors write a review, i.e. a paper discussing the arguments for and against the presence of OSCs in mammalian ovaries in this controversy. Instead, the authors present “first preliminary arguments” (non-verifiable) for “the consistency of OSC population from a single 28 cryopreserved ovarian cortex after thawing”. Once again, if it is a real review, the authors should really and exhaustively discuss why certain labs state that OSCs do exist in the ovaries, and many others have never found any evidence of their existence. Please, tell us how many labs think that these cells exist (and which ones) and how many labs think not?

References are not homogenous. This paper is definitely not seriously written.

Not all the authors:

Zhang, H. et al. Adult human and mouse ovaries lack DDX4-expressing functional oogonial stem cells. Nat. Med. 21, 499 1116– 1118 (2015).

Hernandez, S. F. et al. Characterization of extracellular DDX4- or Ddx4- positive ovarian cells. Nat. Med. 21, 1114–1116 (2015).

Compared to

Truman, AM; Tilly, JL; Woods, DC. Ovarian regeneration: The potential for stem cell contribution in the postnatal ovary to 506 sustained endocrine function. Mol Cell Endocrinol. 2017; 445: 74-84

No title:

Parte, S; Bhartiya, D; Patel, H; Daithankar, V; Chauhan, A; Zaveri, K; Hinduja, I. J Ovarian Res 2014; 25: 7:25 496

White, YAR; Woods, DC; Takai, Y; Ishihara, O; Seki, H; Tilly, JL. White Y.A.R., et al. Nat Med. 2012; 18: 413-421

Abbreviations not homogenous:

Donnez, J; Dolmans, MM; Pellicer, A; Diaz-Garcia, C; Serrano, MS; Schmidt, KT; Ernst, E; Luyckx, V; Andersen, CY. Restoration of ovarian activity and pregnancy after transplantation of cryopreserved ovarian tissue: a review of 60 cases of reimplantation. Fert ster 2013, 1503-1513.

Meirow, D; Fasouliotis, SJ; Nugent, D; Schenker, JG; Gosden, RG; Rutherford, AG. A laparoscopic technique for obtaining ovarian cortical biopsy specimens for fertility conservation in patients with cancer. Fertil steril 1999, 948-951.

Author Response

Reviewer # 2:  In my first comments, I asked that the authors cite and discuss the very many papers stating that they have never found ovogonial stem cells in the ovaries of their mam-malian model. But in the second version of this “review”, the authors contented themselves with citing the articles contradicting the presence of these cells in the ovaries, without discussing them.

First, I would be very interested to discuss with the authors about a quantification of the number of papers that state that OSC exist in comparison with papers that state the contrary.

And I would like to know the arguments of the authors when they state that the other observations are “confuting”. For me they are not all say that OSCs do not exist in mammalian ovaries.

The authors write a review, i.e. a paper discussing the arguments for and against the presence of OSCs in mammalian ovaries in this controversy. Instead, the authors present “first preliminary arguments” (non-verifiable) for “the consistency of OSC population from a single 28 cryopreserved ovarian cortex after thawing”. Once again, if it is a real review, the authors should really and exhaustively discuss why certain labs state that OSCs do exist in the ovaries, and many others have never found any evidence of their existence. Please, tell us how many labs think that these cells exist (and which ones) and how many labs think not?

Answer: These comments probably focus a misunderstanding of the Reviewer with the topic of our article whose content and preliminary data only concern the viability of OSCs after thawing from cryopreserved ovarian cortex fragments. We merely present very early results on both Ddx4 expression and 7-AAD-PC5 incorporation in Ddx4+ cells obtained from cryostored female ovarian cortex. Once again, we would like to emphasize that our article is not devoted to discussing whether or not the OSCs exist with respect to the scientific debate on their widely described or denied subsistence in mammalian ovaries. While we are quite familiar with this intellectual controversy, we belief that such a disagreement helps all of us to persevere our own hypotheses and improve our knowledge in this field of interest, and based on our published work, we do trust in our results. This Reviewer probably belongs to the cohort of deniers of the OSC presence in ovaries and insistently asks to include in the manuscript numbers of citations, articles, labs, research groups and names of researchers as pros and cons to OSCs. All these specifications appear out of the purpose of our article, and we do not believe that would provide knowledge to this scientific topic. However, with the purpose to interpret the Reviewer’s claim, we have now slightly modulated the previous sentence (lines 156-159) describing the number of studies on OSCs, and modified the word “confuting” in “denying” reports (line 156).

References are not homogenous.

References have been now uniformed as requested.

This paper is definitely not seriously written.

We are sorry for this disapproval by the Reviewer. However, we did the best we could in writing a scientific paper.

Reviewer 3 Report

The authors of this study are presenting the idea that ovarian stem cells from cryopreserved ovarian cortex could be a potential source of stem cells/precursor cells, which could potentially be differentiated into germ cells/oocytes in the future. I think that theoretically, the idea is sound, but unfortunately, it is not believed that this will be possible to employ in clinical practice in the next few years. There are several reasons, but the most important is that there is still not a wider consensus about the existence of ovarian stem cells. Even more, there seem to be more possible populations of stem cells present in ovaries. I think that this should be highlighted and shortly describe each putative type of ovarian stem cells. While this review includes also a ''case report'' I suggest this should be mentioned in the title of the manuscript to attract more attention from potential readers (for instance, case report with a literature review or something like that). Also, despite Ddx4 positive cells were isolated, there is still no firm data that they can be differentiated/developed into oocytes therefore this drawback should be more discussed. 

Minor comments:  

  • you use abbreviation OSCs for ovarian stem cells and also for oogonial stem cells. This is not the same
  • page 4, lines158-161: these lines are highlighted with colour. Why?

Author Response

We are grateful for the favorable appreciation of our work and agreeing about a possible employ of the OSCs in clinical practice only in the next years. As required, to emphasize the intellectual debate on OSCs, we have now detailed the studies by all groups sustaining and denying the existence of these cells in mammals. Thus, based on the description of a single isolation of OSCs from frozen ovarian cortex, in agreement with the Reviewer’s suggestion, we have now included in the manuscript title the wording “a case report and review of literature”.

 Finally, we would like to point out that the abbreviations for OSCs now have only been used for ovarian stem cells and not for oogonial and that the lines 158-161 were highlighted for the previous review request.

Round 2

Reviewer 2 Report

Now the authors have added the following sentence to temper comments about the relative number of labs claiming that there are or are not CSOs in the ovaries, “With respect to such a skepticism, the number of studies describing the occurrence of OSCs is perhaps equivalent to the denying reports while their biological properties assessed by independent groups of investigators apparently support that OLCs differentiated by human OSCs are functional cells in vitro though these properties need to be better ascertained for their potential use in vivo.”. And that’s all!!!

Once again, if it is a real review, the authors should really and exhaustively discuss why certain labs state that OSCs do exist in the ovaries, and many others have never found any evidence of their existence. Why do some laboratories (please cite them) recover OSCs with antiDDX4 Antibodies and why others do not? And thank you also for quoting those who say they do not detect them. What antibody did they use? If they are not the same, offer to make exchanges. On the other hand, why has no birth been revealed thanks to the fertilization of "oocytes" from these OSCs? This question is sufficiently important for public health to require some consideration.

Author Response

Once again, our article is not devoted to discussing the existence of OSCs in relation to the scientific debate proving, or denying, the occurrence of these cells in mammalian ovaries. We are familiar with this intellectual controversy and trust that such a disagreement is definitely helpful for us in pursuing our work and improving our knowledge in this fascinating field of interest. However, with the purpose to satisfy the Reviewer’s claim, we have now detailed in the revised manuscript all research groups confirming and denying the existence of OSCs in mammalian ovaries. Furthermore, we also have mentioned the controversial role of Ddx4 marker in isolating the OSCs, as required. To this, we inserted the next sentence that, we hope, may be considered adequate for the Reviewer’s criticism.

To this regard the debate on mammalian neo-oogenesis arose within the scientific community, by several studies published by independent groups of investigators, namely Pacchiarrotti J [28], Bukovsky A [41], Johnson J [42], Virant-Klun I [43], Niikura Y [44], Zou K [45], Pacchiarotti J [46], Parte S [47], Zhang Y [48], and Zhou L [49], sup-porting the existence of OSCs, whereas others including Bristol-Gould SK [34], Liu Y [35], Byskov AG[36], Kerr JB[37], Zhang H[38], Lei L[39] and Yuan Y [40], confuted these results on the basis of technical misundestanding of data and required further evidence for the occurrence of OSCs in all mammals including humans.

In this contest an additional question regards the use of Ddx4 as a germline marker for sorting OSCs [33], which is considered a functional membrane marker for their isolation while its location is apparently detected within the cytoplasm. In reality, it has been reported that the Ddx4 transmembrane-spanning domain is expressed by OSCs in their early maturation step before its setting in a definite intracytoplasmic location in mature oocytes [59,54].

On the other hand, Ddx4+OSCs has been clearly demonstrated to express stemness markers as Oct4 and SSEA4, thus providing evidence in support of active germ cells in adult mouse ovary [33]. The expression of those stemness markers has been confirmed by several methods including detection by specific reagents, RT-PCR and BrdU up-take [60]. Similarly, IFITM3 (Fragilis) has also been found to be a highly specific marker that based on its transmembrane location is functionally utilized for efficient OSC isolation [61]. In fact, Sequeira RC and colleagues in their study proposed a new fruitful isola-tion method combinig anti-Ddx4 and anti-IFITM3 antibodies in MACS sorting, that de-finitive improved the recruitment of OSCs with respect to the method using only san-ti-Ddx4 antibodies [62].

Reviewer 3 Report

As already mentioned in previous review, there are several populations of stem cells present in ovaries, for instance, oogonial, VSELs, mesenchymal etc., and all are not Ddx4 positive as this manuscript is trying to imply at some points. So I suggest just one short correction. In the section where you added new references describing the existence of ovarian stem cells, just add to each reference the type of ovarian stem cells it describes. 

Author Response

We are grateful for the favorable appreciation of our work. As required, now, we have mentioned the several population of stem cells present in ovaries. 

This manuscript is a resubmission of an earlier submission. The following is a list of the peer review reports and author responses from that submission.

Round 1

Reviewer 1 Report

General comment

This paper is not a review but a bias that only presents the story of the pro-Tilly concerning the existence of oogonial stem cells in mammalian ovaries. And who takes the opportunity to slip a case report n=1 without any characterization of the sorted cells (DDX4+) ??? In my opinion ddx4 remains and remains cytoplasmic - although it has been described at the a membrane ???).

Concerning the existence of oogonial stem cells, there is no convincing pregnancy, not even worth rebuttal. In this paper, the authors only cite the groups that agree with J Tilly, and never the very many labs that contradict these results:

Zarate-Garcia, L. et al. FACS-sorted putative oogonial stem cells from the
ovary are neither DDX4-positive nor germ cells. Sci. Rep. 6, 27991 (2016).

Zhang, H. et al. Adult human and mouse ovaries lack DDX4-expressing
functional oogonial stem cells. Nat. Med. 21, 1116–1118 (2015).

Hernandez, S. F. et al. Characterization of extracellular DDX4- or Ddx4-
positive ovarian cells. Nat. Med. 21, 1114–1116 (2015).

The authors cite Wagner et al (Nature communications 2020 11/1147) but not that this paper demonstrates that the cells captured with anti DDX4 antibodies are perivascular cells and not oogonal stem cells. And that oogonial stem cells do not exist in mammalian ovaries.

The authors write "a few skepticisms raised in relation to a 100 foundational dogma in female mammalian reproductive biology [31]", but ref31 is a paper from 1951!

Furthermore, Tilly and Wu groups never share their antibodies with any other group to confirm the whole thing. 

Other comment

- There is no doubt that there are adult stem cells in the ovarian epithelium (as in all epithelia). That stem cells can artificially be brought closer to a pluripotent character is possible (iPS). That pluripotent stem cells can form PGC-LC then oocytes-like is true in vivo and recently in vitro in mice. That we can do everything again one day in a woman yes, why not? But why would you have to look for stem cells that are so difficult to access in the ovarian epithelium (and not in the skin for example), is there any interest? There is the question?

Reviewer 2 Report

The authors described the consistency of the OSC population from a single cryopreserved ovarian cortex after thawing as well as both viability and they explained that interest for stemness studies in the treatment of CTRI and female infertility. This article was well written and could be acceptable for publication. 

Reviewer 3 Report

Congrats. Nice paper. Very important topic which in many cases is still underestimated. Nowadays when we have different ways of action we need to think at least two steps ahead, not only to treat the primary disease but also to have in mind quality of life of our patients.

I have some small remarks to submitted manuscript.

  • there are some editorial/gramatical/spelling mistakes (i.a. 'disconfort') which shuold be easily fixed by revision of the manuscript.
  • multiplying signs should be used instead of "x"
  • on the Fig. 1 use dot instead of coma.
  • I don't understand phrase "the cortical biopsies were performed mostly by laparscopy during the abdominal surgery or specially adopted..." laparoscopy or laparotomy? 
  • I would expect more up-to-date references. There are some important papers missing. Moreover, there are missing data in proper format of cited paper (omitted pages, volumes, etc.) - it also easily should be fixed.

I would recommend publication after minor revision.